# Wave Impact Pressures on Stepped Revetments

**Nils B. Kerpen *** , **Talia Schoonees and Torsten Schlurmann**

Ludwig-Franzius-Institute for Hydraulic, Estuarine and Coastal Engineering, Leibniz University Hannover,
30167 Hannover, Germany; schoonees@lufi.uni-hannover.de (Ta.S.); schlurmann@lufi.uni-hannover.de (To.S.)
* Correspondence: kerpen@lufi.uni-hannover.de; Tel.: +49-511-762-3740

**Abstract:** The wave impacts on horizontal and vertical step fronts of stepped revetments is investigated by means of hydraulic model tests conducted with wave spectra in a wave flume. Wave impacts on revetments with relative step heights of $0.3 < H_{m0}/S_h < 3.5$ and a constant slope of 1:2 are analyzed with respect to (1) the probability distribution of the impacts, (2) the time evolution of impacts including a classification of load cases, and (3) a special distribution of the position of the maximum impact. The validity of the approved log-normal probability distribution for the largest wave impacts is experimentally verified for stepped revetments. The wave impact properties for stepped revetments are compared with those of vertical seawalls, showing that their impact rising times are within the same range. The impact duration for stepped revetments is shorter and decreases with increasing step height. Maximum horizontal wave impact loads are about two times larger than the corresponding maximum vertical wave impact loads. Horizontal and vertical impact loads increase with a decreasing step height. Data are compared with findings from literature for stepped revetments and vertical walls. A prediction formula is provided to calculate the maximum horizontal wave impact at stepped revetments along its vertical axis.

**Keywords:** Stepped revetment; wave impact; physical model test

## 1. Introduction

The increasing population living in coastal areas poses new demands in terms of the environmental and touristic compatibility of coastal protection structures. As aesthetically pleasing coastal protection structures gain increasing importance, and accessible revetments, e.g., stepped revetments become more attractive. Recent installations can be found in Margate, UK (finished 2013), Blackpool, UK (finished 2017), or Chicago shoreline project, US (finished 2018). Furthermore, the stepped surface of a stepped revetment induces additional turbulence in the flow, which leads to increased energy dissipation compared to smooth impermeable structures, however, traditional permeable structures provide more dissipation due to filtration and percolation effects [1,2]. Consequently, wave energy available as kinetic energy for the wave run-up process is reduced. Presently however, practical design guidance is limited for stepped revetments. A comprehensive overview of existing research focusing on the general wave-interaction with stepped revetments is presented in Reference [3]. References [2,4] highlight the reduction capabilities of stepped revetments for wave run-up and overtopping compared to smooth impermeable structures for regular and irregular waves. Reference [5] discusses the energy dissipation within the wave run-up at stepped revetments and detected similarities to steady flow conditions over stepped surfaces like spillways. Thus far, wave impacts on stepped revetments have not been analyzed systematically and are discussed in this study. The paper is structured as follows: First, the current understanding of wave impacts on stepped revetments is presented in Section 2 and key knowledge gaps outlined. Section 3 describes the geometrical and hydraulic boundary conditions of the conducted model tests. The analysis and

discussion of the results for wave impact pressures with special focus on (1) the probability distribution, (2) the time evolution including load case definitions, and (3) the specific evolution of wave-induced pressure is given in Section 4. Section 5 focuses on the underlying laboratory and scale effects in the physical model. Finally, the results are summed-up and contrasted in context of the present state-of-the-art and the advancement of knowledge outlined through this new study.

## 2. Previous Studies on Wave Impacts on Stepped Revetments

Knowledge of loads on a coastal protection system is essential to determine its final design and performance over its lifetime. Hydraulic loads can be classified as either hydrostatic or hydrodynamic. Only the hydrodynamic load cases are addressed in this paper.

A general classification of breaking wave loads on vertical structures for different breaker types is given by Reference [6]. Dynamic impact loads most often occur due to plunging waves impacts on a marine structure. The progress of a single wave impact is dependent on the hydraulic- and geometry-related boundary conditions (described in this section). A stepped revetment has horizontally and vertically aligned fronts in relation to the direction of gravity. The loads on these two fronts are different due to the asymmetry and phase-shift of orbital velocities in shallow waters and the asymmetric structure of the wave-form represented by the wave steepness (wave height $H \ll$ wave length $L$). Furthermore, the effect of wave loads is dependent on the overall slope ($n$) of a structure. A unique loading case is represented by a plain vertical wall without horizontal planes where an overturning, i.e., breaking wave crest violently slams against the wall without any damping or dissipation due to preceding waves. The other extreme case is a very gentle slope (nearly horizontal) where plunging and spilling breaking waves run-up on an inclined, often impermeable plane (dependent on the Iribarren number $\xi = \tan \alpha / \sqrt{H/L}$). In the case of spilling breaking waves the exposed load on the structure is regularly damped by previous waves. On a structure with horizontal and vertical fronts, tongues, and jets of water can be created from the breaking wave and be directed upwards like an up-rushing jet of water [1]. This jet of water instantaneously collapses and falls downwards, and in turn, induces an additional impact on the horizontal fronts. It is evident that this loading on a horizontally aligned front (tread of the step) is smaller than on a vertical front (riser of the step).

Recent literature provides only little insight on wave impacts on stepped revetments. According to Reference [7], the forces on stepped structures should be calculated for design purposes with the same method as for vertical walls since the dynamic pressures are in the same range. Reference [8] conducted hydraulic model tests with irregular waves for a stepped seawall in a 76 m long wave flume following a Froude scale of 1:19. These tests focused on measuring the reduction of wave overtopping and the wave-induced impacts on stepped seawalls. Two sloping structures ($n = 1.5$, $n = 2$) with step heights of $S_h = 0.026$ m were analyzed. A recurved seawall was incorporated at the crest of both sloping structures. Reference [8] analyzed the wave loads on stepped structures in a specific range of Iribarren numbers of $2.8 < \xi < 6.3$ and step ratios in a range of $9.0 < H_{m0}/S_h < 11.0$ with $H_{m0}$ as zeroth moment wave height. Similarly, Reference [8] remarks the importance of the short duration shock pressures (impacts) resulting from the rapid compression of an air pocket trapped between the front of a breaking wave and the wall. For vertical walls the authors in Reference [6] note that 'the shock pressure exerted by a breaking wave is due to the violent simultaneous retardation of a certain limited mass of water that is brought to rest by the action of a thin cushion of air, which in the process becomes compressed by the advancing wave front' [9]. The position of the highest measured impacts was dependent on the initial still water level (SWL). According to the analysis of impact distributions, the maximum impacts at different wall elevations rarely occur simultaneously. This finding is particularly valid in the case of a non-vertical wall, such as the stepped wall studied here, since some wave energy is dissipated through turbulence. In some cases, a negative impact duration was measured, which is interpreted as a characteristic of turbulence and air entrainment occurring at the base of each seawall step. Finally, Reference [8] summarizes a discussion about the importance of shock pressures for the actual design

of a stepped seawall. According to the discussion, pressures of such short duration should not be used for establishing the design load case. Rather, it is recommended to consider the smaller surge pressures with a longer duration to determine the critical dynamic load.

In addition, Reference [10] conducted hydraulic model tests in a Froude scale of 1:20 for stepped revetments ($S_h$ = 0.015 m), focusing on wave run-up and wave overtopping. The vertical wave impact was measured on a single step with a sampling rate of only 100 Hz, which is considered as rather inappropriate for measuring rapid, i.e., almost instantaneous, wave impact pressures. As such, analyzed data represent merely averaged maximum impact pressures $P_{max}$ of six test repetitions (with a standard deviation of $STD{\sim}0.1\,P_{max}$) to depict the inherent loading bias within each experiment.

Both studies lack a comprehensive discussion on the instantaneous pressure impact events with corresponding wave conditions. Hence, the systematic analysis of the wave impacts on stepped revetments conducted in the experiments in the present study includes a comparison with the data and evaluations provided by References [8,10].

## 3. Experimental Set-Up, Test Conditions and Procedures

Hydraulic model tests focusing on the wave interaction with stepped revetments were conducted in a wave flume, which has a length of 110 m, a width of 2.2 m and an overall depth of 2.0 m. For these tests, wave spectra were calculated with second order wave theory routines. The irregular wave profiles ($H_{1/3,max}$ = 0.42 m with $T_{max}$ = 2.0 s) were generated with a piston-type wave maker. Two model set-ups, constructed from plywood, with varying step heights (large steps: $S_h$ = 0.3 m, small steps: $S_h$ = 0.05 m) are placed in a 0.7 m wide sub section over a horizontal flume bottom at a distance of $L_F$ = 81.6 m from the wave paddle (Figure 1). The relative flume length with respect to the tested wave length $L_p$ is $10 < L_F/L_p < 36$.

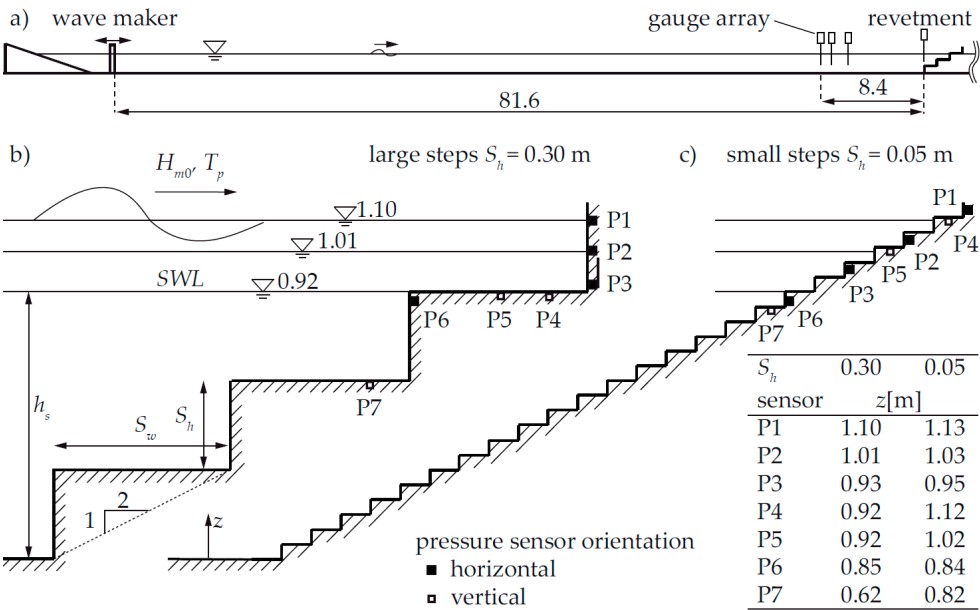

**Figure 1.** Model set-up of pressure sensors and position of the *SWL* for two 1:2 inclined stepped revetments. (**a**) Cross-section of the flume set-up, (**b**) detail of the large step configuration ($S_h$ = 0.3 m), and (**c**) detail of the small step configuration ($S_h$ = 0.05 m).

The surface elevation is measured by five ultrasonic sensors with a measuring range of 200 to 1200 mm, a superior resolution of 0.36 mm and a sampling rate $f_{sample.}$ = 50 Hz. Three of the sensors are positioned at a distance larger than two wave lengths $L$ from the toe of the revetment, to determine incident wave conditions, as calculated by a reflection analysis. One sensor is placed at the toe of the stepped revetment and another in the shallow water region of the still water level. Pressure impacts

on the stepped revetment are recorded by seven pressure transducers, which are placed along the horizontally ($f_{sample.}$ = 2.4 kHz) and vertically ($f_{sample.}$ = 19.2 kHz) orientated step fronts (Figure 1). An impression of the set-ups is given in Figure 2 for the analyzed step heights of 0.05 m (a) and 0.3 m (b). In order to capture a profile of wave induced loads on an 1:2 inclined stepped revetment, sensor locations are varied in relative water depths —6.0 < $z/H_{m0}$ < 2.0, relative to the still water level. The pressure sensors (ATM.1ST/N fabricated by sts-sensors) have a range from zero to 150 mbar and a non-conformity of ±0.1% from full scale. The sensors are connected by a serial interface connection (RS232) to the data acquisition and provide an output signal from zero to 10 V. The configuration of the probes allows a local (over a single step) and a more global interpretation (for the whole revetment).

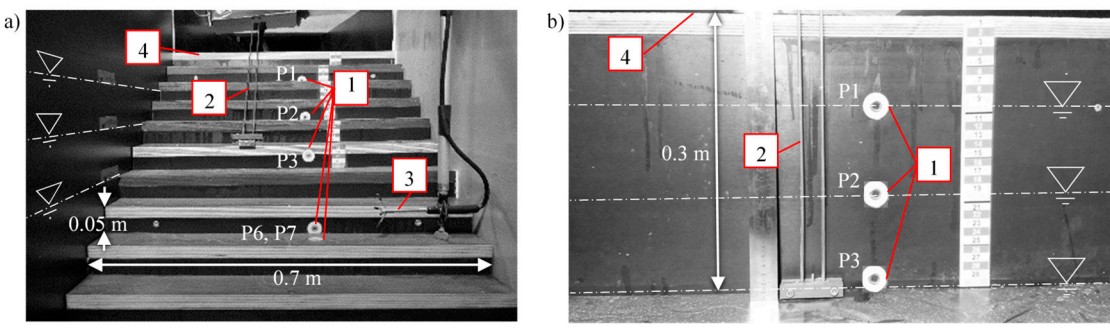

1 Pressure sensors　　2 Run-up gauge　　3 ADV　　4 Crest level

**Figure 2.** Model set-up with instrumentation for (**a**) step height 0.05 m and (**b**) step height 0.3 m.

The hydraulic boundary conditions of the wave impact experiments are listed in Table 1. The parameter choice covers a wide range of dimensionless variables with a wave steepness of 0.015 < $H_{m0}/L_p$ < 0.04 (with $H_{m0}$: spectral wave height, $L_p$: wave length calculated from the peak period $T_p$ measured at the gauge array in a distance of $x$ = 8.6 m (1 $L_p$ < $x$ < 3.8 $L_p$) from the toe of the revetment) and Iribarren numbers of 2.5 < ξ < 4.9. Three different water levels $h_s$ with intermediate water depths (0.13 < $h_s/L_p$ < 0.49) are tested. Additionally, the corresponding freeboard height $R_c$ and the number of waves $N$ in each test are given. A total of 13 tests are conducted for steps with a height of $S_h$ = 0.05 m (1.1 < $H_{m0}/S_h$ < 2.8) and 10 tests with a step height of $S_h$ = 0.3 m (0.2 < $H_{m0}/S_h$ < 0.6).

**Table 1.** Hydraulic boundary conditions for the wave impact tests.

| Test | $S_h$ | $R_c$ | $H_{m0}$ | $T_p$ | $h_s$ | $N$ | ξ | $H_{m0}/L_p$ | $H_{m0}/h_s$ | $H_{m0}/S_h$ |
|------|-------|-------|----------|-------|-------|-----|---|--------------|--------------|--------------|
| # | (m) | (m) | (m) | (s) | (m) | (-) | (-) | (-) | (-) | (-) |
| 101 | 0.05 | 0.121 | 0.056 | 1.43 | 1.100 | 1298 | 3.7 | 0.018 | 0.051 | 1.12 |
| 102 | | | 0.063 | 1.20 | 1.100 | 427 | 3.0 | 0.028 | 0.057 | 1.26 |
| 103 | | | 0.084 | 1.38 | 1.100 | 1261 | 2.9 | 0.029 | 0.076 | 1.68 |
| 104 | | | 0.082 | 1.38 | 1.100 | 1256 | 3.0 | 0.028 | 0.074 | 1.63 |
| 105 | | | 0.082 | 1.38 | 1.100 | 1261 | 3.0 | 0.028 | 0.075 | 1.64 |
| 106 | | | 0.084 | 1.37 | 1.100 | 167 | 2.9 | 0.029 | 0.076 | 1.67 |
| 107 | | | 0.088 | 2.11 | 1.100 | 1422 | 4.2 | 0.015 | 0.080 | 1.76 |
| 108 | | | 0.114 | 2.20 | 1.100 | 171 | 3.8 | 0.019 | 0.104 | 2.28 |
| 109 | | | 0.119 | 2.81 | 1.100 | 179 | 4.6 | 0.016 | 0.108 | 2.38 |
| 110 | | | 0.143 | 2.26 | 1.100 | 162 | 3.5 | 0.023 | 0.130 | 2.86 |
| 111 | | 0.211 | 0.085 | 2.09 | 1.010 | 1410 | 4.2 | 0.016 | 0.084 | 1.71 |
| 112 | | | 0.085 | 2.07 | 1.010 | 1434 | 4.1 | 0.016 | 0.084 | 1.70 |
| 113 | | | 0.085 | 2.08 | 1.010 | 1371 | 4.1 | 0.016 | 0.084 | 1.70 |
| 201 | 0.30 | 0.121 | 0.111 | 1.37 | 1.100 | 1413 | 2.6 | 0.038 | 0.101 | 0.37 |
| 202 | | | 0.129 | 3.18 | 1.100 | 307 | 4.9 | 0.016 | 0.117 | 0.43 |
| 203 | | | 0.116 | 1.38 | 1.100 | 1428 | 2.5 | 0.040 | 0.106 | 0.39 |
| 204 | | | 0.167 | 2.08 | 1.100 | 278 | 3.0 | 0.030 | 0.151 | 0.56 |
| 205 | | 0.211 | 0.064 | 1.36 | 1.010 | 1586 | 3.3 | 0.023 | 0.064 | 0.21 |
| 206 | | | 0.091 | 1.34 | 1.010 | 1390 | 2.8 | 0.033 | 0.090 | 0.30 |
| 207 | | | 0.166 | 2.01 | 1.010 | 1515 | 2.9 | 0.032 | 0.164 | 0.55 |
| 208 | | 0.300 | 0.064 | 1.41 | 0.921 | 1339 | 3.4 | 0.021 | 0.069 | 0.21 |
| 209 | | | 0.089 | 1.38 | 0.921 | 1294 | 2.8 | 0.031 | 0.097 | 0.30 |
| 210 | | | 0.170 | 2.11 | 0.921 | 1468 | 2.9 | 0.032 | 0.184 | 0.57 |

Raw data from the pressure sensors are offset-corrected by means of the first five seconds of the data. The time and amplitude of the peaks are calculated for minimum peak heights of 10 mbar and a minimum distance between two peaks of 0.8 times the wave period $T_p$.

Every single wave in a wave spectrum causes an individual impact $p$ on the structure. The magnitude of the single impact depends on its individual wave kinematics and the influence of the previous wave (remaining water layer over a pressure sensor and amount of aeration in the wave). The analysis of wave loads includes a number of parameters such as the maximum impact pressure $P_{max}$. The quantity of a finite number $N$ of waves that cause $N$ individual impacts $p$ within a single test is defined as $P$. If the quantity $P$ is sorted in a descending order, the maximum recorded impact $P_{max}$ is defined as max$\{P\}$ or $P_{(i=1)}$, as given in Equations (1) and (2).

$$P = \{p_1, p_2, p_3, \cdots, p_N\} \quad with: \ p_1 > p_2 >, p_3 > \cdots > p_N \tag{1}$$

$$P_{max} = \max\{P\} = P_{(i=1)} = p_1 \tag{2}$$

While the maximum induced impact $P_{max}$ is very important for the design of a structure, it presents significant scatter when formulating predictions for practical design purposes. Therefore, the collected pressure magnitudes will rather be described with a probability of exceedance (e.g., 2% of all incident waves reveals the probability of exceedance impact pressure $P_{98\%}$, Equation (3)), which offers more representative and reliable predictions.

$$P_{98\%} = \frac{\sum_{i=1}^{n} p_i}{n} \quad with: \ n = N(1 - 0.98) \in \mathbb{N} \tag{3}$$

In order to compare the measured data to data from previous investigations (e.g., Reference [11]), the impact magnitudes exceeded by the four highest waves out of a 1000-wave test should be calculated. Hence, $P_{99.6\%}$ is calculated on the basis of the number of waves ($N$) given in Table 1. The impact with a certain probability of exceedance ($P_{99.6\%}$ in this case) is calculated finally by the mean of the $n$ highest impact events according to Equation (4).

$$P_{99.6\%} = \frac{\sum_{i=1}^{n} p_i}{n} \quad with: \ n = N(1 - 0.996) \in \mathbb{N} \tag{4}$$

## 4. Results and Discussion

The general differences in the wave impact and wave run-up for a plain slope and stepped revetments with two different step ratios are given in Figure 3 for a better comparison and understanding of the results presented in this section. The color-scheme of the initial black and white images recorded with a frame rate of 50 Hz is inverted for the sake of visual clarity. Small tracers (tiny black dots in the images) that follow the main flow field induced by the up-rushing wave are visible on each image. Intense dark areas represent a high amount of aeration due to wave breaking and wave-structure interaction, as such, a corresponding high flow velocity. Figure 3 shows the wave-induced processes on a 1:2 sloped revetment with smooth impermeable surface in chronological order, i.e., the wave impact (a), the wave run-up (b,c), and finally, the wave run-down (d) of a breaking wave. The time step $t$ of each frame is given in relation to the wave period $T$. In analogy, Figure 3e–h shows the wave impact and wave run-up of a wave with the same boundary conditions over a revetment with small steps ($H/S_h$ = 1.7) and Figure 3i–l over a revetment with large steps ($H/S_h$ = 0.3). For the two stepped configurations, the still water level *SLW* is at the level of a step edge.

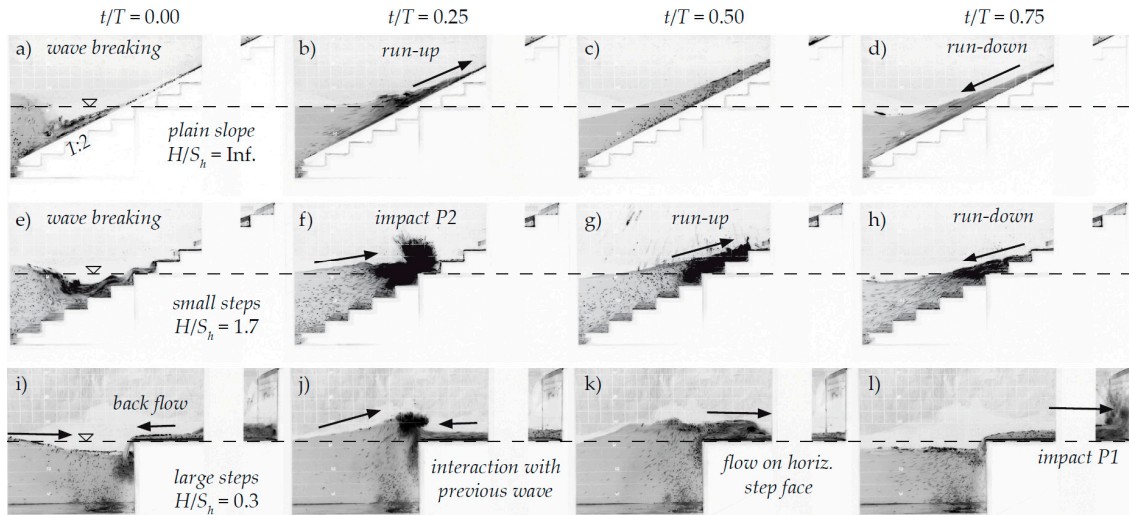

**Figure 3.** Demonstration of flow processes for relative time-steps in the run-up process on revetments with smooth surface (**a–d**), small step heights (**e–h**), and large step heights (**i–l**).

On a smooth plain slope, a breaking wave induces air in the water body. The wave run-up (b) contains water and air. At the maximum point of wave run-up excursion (c) the entire fraction of air is leaked from the water body, and in consequence, the wave run-down (d) contains only water. During the wave run-down the preceding wave then interacts with the next incoming wave and the described run-up and aeration process is repeated. For revetments with small steps ($H/S_h = 1.7$), the general flow processes of wave breaking, wave run-up and wave run down sequence is almost in analogy with the processes on a smooth slope. The main differences are: (1) The impact of the breaking wave causes a splash-up at the vertical step front (f) during the wave run-up process, (2) the aeration during the wave run-up is more intense during the run-up process caused by larger friction induced by the steps while interacting with the incoming waves, and (3) higher turbulence and continuous air intrusion (g). Yet, in contrast, the wave run-down induces air into the water body and is retarded compared to the smooth revetment (h). For large step heights ($H/S_h = 0.3$), the flow processes differ significantly from a normal wave run-up as shown in (a–h). The incident breaking wave interacts with the reflected backflow of a previous wave at the first step edge (i). The interaction leads to an impact on this first step edge, an upwards-directed flow and a wave breaking in very shallow waters over the step edge (j). Then, the wave is propagating as spilling breaker over the horizontal step front (k) until it impacts and resonates with the second vertical step front (l). The impact induces a second splash up.

The previous described processes during the wave run-up over a plain and stepped revetment resemble the major dissimilarities, namely (1) the higher volume of aeration, (2) the stronger interaction with and the influence of previous waves, and (3) the higher breaking wave impact characteristic (impact direct on the slope/step front, impact on a water cushion induced by the previous wave). In summary, these previously outlined effects feature the structure and functioning of stepped revetment by means of reflecting and dissipating incoming wave energy on a sloping beach leading to less wave run-up and, in consequence, to lower wave overtopping rates. In order to quantify the influence of the formerly described processes, the induced wave impact pressures on stepped revetments are analyzed. This examination focuses on (1) the probability distribution of impact pressures, (2) the temporal evolution and typical load cases of impact pressures, and (3) the spatial distribution of impact pressures. Attention is drawn to these three foci in reference to the step ratio. Recent literature on wave impacts on impermeable plain slopes, stepped revetments and vertical seawalls are presented to complement the findings of this research and ensure a comprehensive analysis of processes.

### 4.1. Probability Distribution of Impact Pressures

According to References [12,13], the probability distributions of wave impact pressures can be described in agreement with mathematical fitting approaches based log-normal functions. Additionally, it has been argued that these log-normal functions are valid in model scale and prototype.

Hence, Figure 4 gives the log-normal probability distribution over the maximum impact pressure for two exemplary tests (test number 103 and 209 according to Table 1). The tests were conducted with a spectral wave height of $H_{m0}$ = 0.084 m and $H_{m0}$ = 0.089 m (for test 103 and 209, respectively) and a corresponding Iribarren number of $\xi$ = 2.9 (2.8 for test 209). Figure 4a provides measurements of wave impacts on slopes with a small step height ($H_{m0}/S_h$ = 1.68, $h_s$ = 1.1 m, sensor *P1*). A total of 876 individual wave impacts have been recorded with pressure sensor *P1* with a maximum recorded pressure impact of $P_{max}$ = 1.26 kPa. The wave impacts are presented with a certain probability of exceedance. The figure presents an idealized normal-distribution, shown as a dashed line, as reference. As is evident from the presented figure, the wave impacts on this stepped revetment follow a log-normal distribution. Over and under predictions from the idealized normal-distribution given with the dashed line appear for impacts larger than $P_{95\%}$ and smaller than $P_{10\%}$.

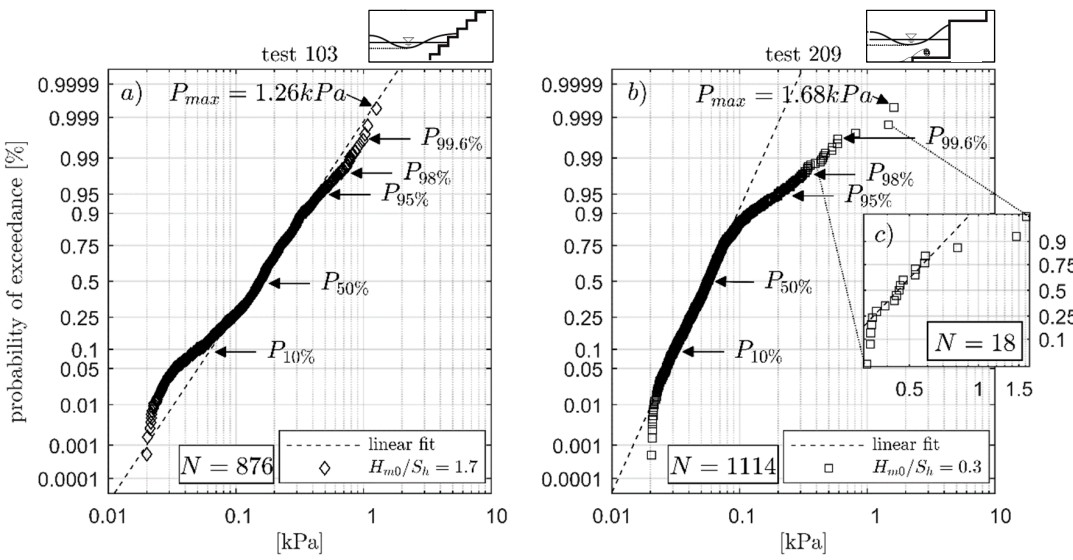

**Figure 4.** Recorded pressure impacts and log-normal probability distributions of the maximum impact pressures for (**a**) small step heights (test 103, pressure sensor *P1*, $H_{m0}/S_h$ = 1.7) and (**b**) large step heights (test 209, pressure sensor *P2*, $H_{m0}/S_h$ = 0.3). (**c**) Gives a detail of the log-normal probability distribution of the 18 largest impacts in test 209.

Figure 4b presents the log-normal distribution of wave impact pressures on a revetment with large steps ($H_{m0}/S_h$ = 0.3, $h_s$ = 0.921 m, sensor *P2*) for the same hydraulic boundary conditions, as in Figure 4a. The maximum pressure impact $P_{max}$ of 1114 individual impacts is 1.68 kPa, which is about 35% larger when compared to the small steps. As for Figure 3c, slight deviations from the ideal log-normal distribution are detected for impacts smaller than $P_{10\%}$, while impacts larger than $P_{90\%}$ deviate significantly from this trend. These deviations indicate that the influence of the wave-interaction with large steps differs for high and low impact scenarios. The apparent difference in the wave breaking and wave run-up at these large steps compared to slopes with and without smaller steps (Figure 3) is a significant transformation of the incident wave which breaks over the very shallow horizontal step front. Hence, the log-normal distribution of the induced pressures of incident wave heights is disturbed and transformed over large steps directly at the structure, and in analogy, the wave impact distribution is affected. The increase in the wave impacts for large steps can be explained by the fact that the vertical step fronts fall completely dry during the run-down process, whereas for small

step heights, there is always a thin water flow remaining from the previous run-down. This thin water layer (often aerated) reduces the wave impact of the next breaking wave.

In parallel, the geometry of a wave front has a significant influence on its resulting wave impact. Generally, the wave height plays a dominant role. It is observed, that small wave heights in the spectrum lead to pulsating extreme loads, whereas the largest waves lead to impacting loads. Distribution of wave impacts deviate strongly from the log-normal distribution for less frequent waves indicating occasional extreme loads on the structure, which is most relevant for practical design attempts. Reference [14] indicated that for vertical walls impacts smaller than 0.4 $P_{max}$ are based on pulsating loads and larger impacts on impacting loads. Reference [15] found that often only a part of all impacts show a good agreement with the log-normal distribution according to the stochastic pattern of the pressures in the impact area. This observation is confirmed for the present experimental configuration. In contrast, when re-considering only the highest impacts during one wave impact event on the area, it turns out that measured data points follow an almost linear relation in a log-normal plot. Hence, Figure 4c gives a detailed log-normal probability distribution of only the 18 largest impacts (>$P_{98\%}$) of test 209, confirming evaluations stemming from Reference [15] to be also valid for stepped revetments.

It can be seen from the left-hand side of this diagram that only a part of them show good agreement with the log-normal function according to the stochastic pattern of the pressures in the impact area. However, if the highest pressure on the area during one wave impact event is selected and plotted in log-normal scale the measured points nearly fall on a straight line which verifies the log-normal distribution.

### 4.2. Temporal Evolution and Load Cases of Impact Pressures

The temporal evolution of wave impact pressures on vertical structures was initially categorized by Reference [6] and analyzed in depth by Reference [16]. According to Reference [16], impacting loads on vertical structures are defined as impact peaks that are minimum 2.5 times larger than the subsequent quasi-static peak $P_q$. Partially breaking waves induce peaks in a range of $1.0 < P_{max}/P_q < 2.5$. Pulsating impacts due to standing waves occur when the impact peak and its subsequent quasi-static peak $P_q$ are in the same order of magnitude. Figure 5 sketches a parameter definition describing both a pressure impact event (a), and an exemplary time series of the maximum impact event during test 204 for horizontal oriented pressure sensors *P1*, *P2*, and *P3* (b). A typical time evolution with an impact and subsequent quasi-static peak is observed. The highest impact is recorded by sensor *P2*, which is located near the still water level *SWL*. The oscillation of the signal and negative pressures, as also observed by Reference [8], are interpreted as characteristically mimicking physical processes of turbulence and air entrainment occurring at the base of each revetment step. With increasing distance to the *SWL*, the impact amplitude decreases (*P1* and *P3*). While sensor *P2* recorded a violently impacting load, *P1* was impacted by a slightly breaking wave and *P3* experienced a pulsating impact.

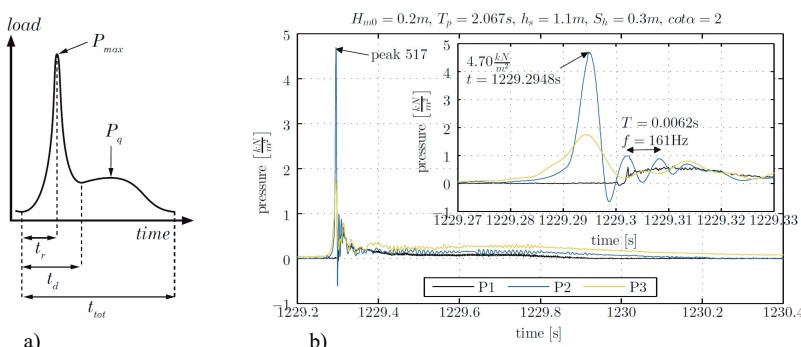

**Figure 5.** (**a**) Parameter definition describing a pressure impact event and (**b**) exemplary time series of the maximum impact event during test 204 for pressure sensors *P1*, *P2* and *P3*.

Figure 6 presents the normalized time evolution of impact events with a certain probability of exceedance for a 1:2 stepped slope with large (test 103) and small step heights (test 209). These tests were conducted with a similar spectral wave height and wave steepness (Table 1). Raw data (black line) are given to indicate the largest impact. A filtered time series (red line, 6th order low pass filter with 38.4 Hz cut-off frequency) is superimposed in order to highlight a clearer temporal progress. Subfigures (a–f) correspond to step ratios of $H_{m0}/S_h = 1.7$ (small steps) and subfigures (g–l) to step ratios of $H_{m0}/S_h = 0.3$ (large steps). Each subfigure displays the measured impact event of a certain probability of exceedance ($P_{max}$ to $P_{10\%}$) over the relative time. The time $t$ is normalized by the wave period $T_{m-1,0}$. Time step $t/T = 0$ is set at the time of maximum impact. Each peak impact (based on the raw data) is given proportionally to the maximum measured peak $P_{max}$ of the entire test duration.

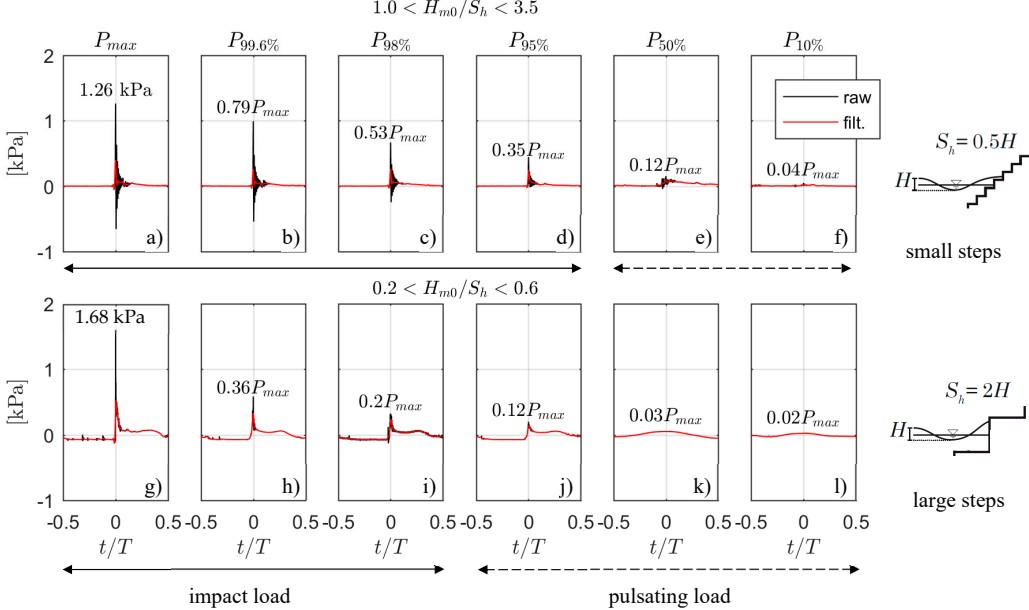

**Figure 6.** Comparison of pressure events with a certain probability of exceedance between small steps ((**a–f**), $H_{m0}/S_h = 1.7$, test 103) and large steps ((**g–l**), $H_{m0}/S_h = 0.3$, test 209).

The maximum measured impact for the small steps (a) was 1.26 kPa, which is about 30% lower than the maximum impact for large steps (g). For the large steps the maximum peak amplitude decays more rapidly with increasing probability of exceedance than for the small steps. In the case with smaller steps, a peak amplitude of about 35% of the maximum impact $P_{max}$ is reached by 5% of all impact events ($P_{95\%}$) (d), whereas in the case of the large steps (h), it is only 0.4% ($P_{99.6\%}$) of all impact events.

A comparison of the occurring load cases defined by Reference [16] elucidates that large steps ($H_{m0} < S_h$) demonstrate a behavioral function like vertical walls in reflecting incoming wave energy. Functional processes are mimicked in two distinct phases: A clear initial impact followed by a quasi-static peak $P_q$. For small steps ($H_{m0} > S_h$) the impact peak is also clearly visible, but the subsequent quasi-static peak $P_q$ is not as prominent as for large steps or vertical walls. These differences are caused due to different flow principles. At large steps, the water level in front of the step front is constantly rising after the initial impact, thus inducing the quasi-static load. At small steps, which dissipate energy from the up-rushing wave tongue, a highly aerated flow emerges after the initial impact (Figure 3f,g) and the absolute depth under the wave crest is evidently lower. This leads in general to smaller quasi-static peaks. For the specific case presented in Figure 6, impacting loads are observed in the range of about 5% of all impacts at the small steps and 2% of all impacts at the large steps. The initial violent impact dissipates more energy than a pulsating load induced by wave run-up. As a result, it can thus be deduced that the overall energy dissipation at small step heights is larger, i.e.,

more effective. Furthermore, the impact-mitigating effect due to the practically permanent existence of a thin water layer on top of the individual small steps of the revetment during the wave run-down phase stemming from the preceding wave (Figure 3) plays a dominant role in impact reduction.

The characteristics of an impact pressure on stepped revetments can be described by the rising time $t_r$ and impact pressure duration $t_d$, as defined in Figure 7 and depicts the correlation between the dimensionless impact duration $t_d/t_r$ and the dimensionless impact rising time $t_r/T_{m-1,0}$. The data are grouped in terms of the step ratios $H_{m0}/S_h$, the direction of the measured wave impact (horizontal or vertical with respect to gravity) and the position of the impact (above or below the still water level *SWL*). As a meaningful reference, Figure 7 presents the empirical projections for rising times and durations of impacts at vertical walls provided by Reference [16]. Impacting load cases ($P_{max} > 2.5$ $P_q$) are characterized by a short relative rising time ($0.02 < t_r/T_{m-10} < 0.2$), while pulsating loads are distinguished by their longer peak durations ($t_r/T_{m-10} > 0.2$). The dominance of a peak is characterized by the relative impact duration $t_d/t_r$. The highest impact loads (e.g., $P_{max}$ or $P_{99.6\%}$) correspond to short relative impact durations $t_d/t_r$ in combination with a short relative rising time $t_r/T_{m-10}$. The larger the relative impact duration $t_d/t_r$, the more critical the impact gets in terms of causing damage or instability of the revetment.

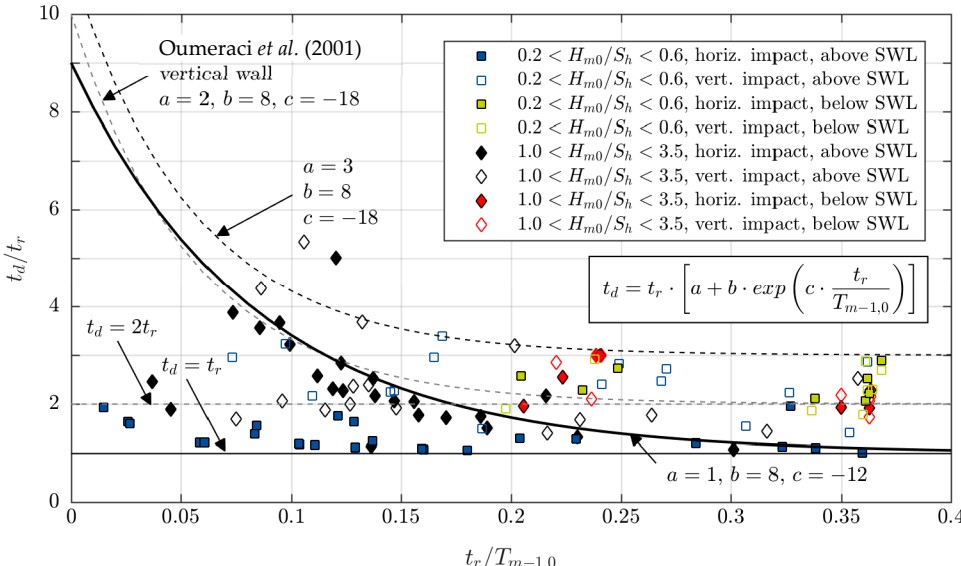

**Figure 7.** Correlation of the dimensionless impact duration $t_d/t_r$ and the dimensionless impact rising time $t_r/T_{m-1,0}$ at stepped revetments compared to findings for vertical walls. Given trend lines following Equation (5) represent upper envelopes of the findings. Straight horizontal lines represent the corresponding lower envelope line.

Impacts measured on stepped revetments below the *SWL* (green and red markers) show relative rising times of $t_r/T_{m-10} > 0.2$. Horizontal impacts (filled marker) or vertical impacts (empty marker) have comparable impact rising times and impact durations. Below the *SWL*, no impacting load case was detected for either small or large steps. The recorded loads have a pure hydrostatic nature induced due to the water level changes induced by wave run-up and run-down over the pressure sensors. Stronger impacts are mitigated by means of a sufficient water layer ("cushion") effectively sheltering the submerged steps' fronts from more violent impacts. The stepped shape of the slope retards the wave run-down leading to a permanent water cover of the revetment below the *SWL*.

The relative impact rising time of horizontal impacts above the *SWL* (filled blue squares for large steps $0.2 < H_{m0}/S_h < 0.6$ and filled black diamonds for small steps $1.0 < H_{m0}/S_h < 3.5$) ranges from very short impacts ($t_r/T_{m-10} < 0.05$) up to long load cases ($t_r/T_{m-10} > 0.3$). The latter ones represent a full run-up and run-down phase inducing pulsating loads. While the impacts at small steps show an

increase in the peak duration for decreasing relative peak rising times up to $t_d = 4t_r$, the maximum duration of $t_d = 2t_r$ is observed for large steps. This finding can be explained with the fact that impacts at large steps are often caused by breaking waves that directly hit the step front, whereas for small steps, the step fronts are covered by thin water layers. Rising times and peak durations of vertical impacts above SWL (empty blue squares for large steps $0.2 < H_{m0}/S_h < 0.6$) and empty black diamonds for small steps $1.0 < H_{m0}/S_h < 3.5$) scatter significantly and do not follow a clear, yet visual trend. The formation and progression of an aerated water layer over the horizontal step front influences the recorded impact significantly and explains the scattering in the data. Pulsating loads can be explained by the water layer thickness during the wave run-up and run-down. Impacting conditions occur only very close to the SWL. The impact rising times for vertical walls and stepped revetments are in the same range. Reference [16] found a minimum peak duration of $t_d = 2t_r$ for vertical walls, whereas the minima for stepped revetments is $t_d = t_r$. These differences may be due to the differences in the sampling frequency of the impact pressure sensors (0.6 to 1.0 kHz at [16] and 19.2 kHz in the present study). Lines of best fit for the different load cases are calculated according to Equation (5) with corresponding regression coefficients $a$, $b$ and $c$ given in Table 2.

$$t_{d,max} = t_r \cdot \left[ a + b \cdot exp\left( c \cdot \frac{t_r}{T_{m-1,0}} \right) \right] \tag{5}$$

**Table 2.** Coefficients $a$, $b$ and $c$ for Equation (5) and corresponding minimal impact duration $t_{d,min}$.

| Geometry | Impact Direction | $a$ | $b$ | $c$ | $t_{d,min}$ |
|---|---|---|---|---|---|
| Vertical walls [16] | horizontal | 2 | 8 | −18 | $2.0\,t_r$ |
| Stepped revetments | horizontal | 1 | 8 | −12 | $1.0\,t_r$ |
| | vertical | 3 | 8 | −18 | $1.5\,t_r$ |

### 4.3. Spatial Distribution of Impact Pressures

For sloping structures the maximum wave impact resulting from depth-induced wave breaking occurs slightly below the SWL [12]. This section presents the maximum wave impacts on stepped revetments at different locations relative to the SWL. Figure 8 gives the horizontal (a) and vertical (b) relative maximum pressure impact distribution over large (dashed, Test 209) and small (solid, Test 103) stepped revetments. For each pressure sensor $P_j$, the maximum impact $P_{j,max}$ is normalized by the maximum pressure impact of all compared sensors ($P_{max} = \max\{P_{j,max}\}$). For this case, the absolute maximum was recorded by sensor $P2$ of the revetment with the small steps (compare Figure 1).

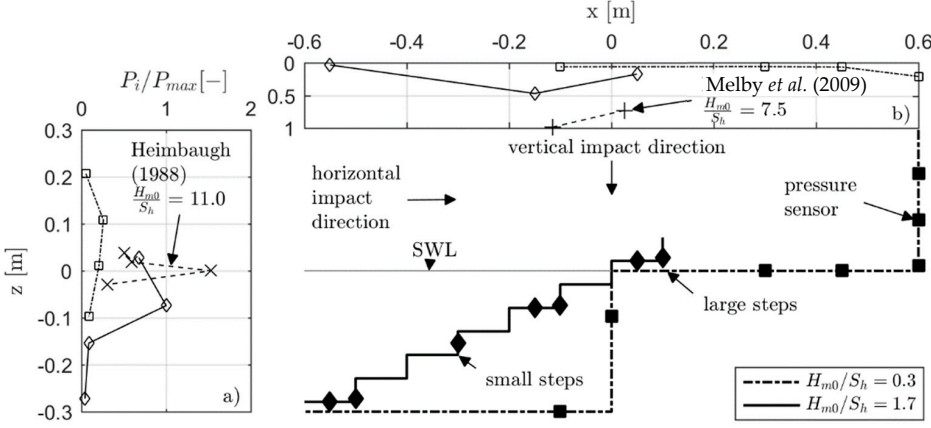

**Figure 8.** Horizontal (**a**) and vertical (**b**) relative maximum pressure impact distribution over large (dashed, Test 209, $H_{m0} = 0.089$ m, and $\xi = 2.8$) and small (solid, Test 103, $H_{m0} = 0.084$ m, and $\xi = 2.9$) stepped revetments.

The vertical distribution of horizontal wave impacts (Figure 8a) shows that the maximum wave impact for all configurations occurs close to the *SWL*. The impact loads decrease with increasing distance to the *SWL* ($|z| > 0$). Below the *SWL*, the pressures are reduced by the presence of the water body, which mitigate the impact of a breaking wave. Above the *SWL* the energy of the breaking wave is less due to the decrease of the potential energy (reduced drop height). Impacts above the SLW are often caused by the secondary impact of the breaking wave during the wave run-up or due to splashes induced by the primary wave impact close the *SWL*. For Test 209 the *SWL* is directly located at the edge of the large step ($H_{m0}/S_h$ = 0.3), while the maximum horizontal impact is about four times lower than at the small steps (Test 103, $H_{m0}/S_h$ = 1.7). Additionally, data from Reference [8] are shown. These data represent very small step heights ($H_{m0}/S_h$ = 11) and indicate a maximum horizontal wave impact of about 1.5 times larger than for the steps with a step ratio of $H_{m0}/S_h$ = 1.7. The smaller the step height in relation to the wave height, the higher the recorded horizontal loads become. The reduced impact for an increasing step height can be ascribed to the delayed run-down of the previous wave impact (compare Figure 3a,e). This run-down causes a constant water layer on the revetment, which buffers the wave impact. Moreover, the plunging wave breaking at a 1:2 slope with small steps ($H_{m0}/S_h \geq 1.7$) triggers impacting wave loads whereas the transformation of the breaking wave by large dominant step edges ($H_{m0}/S_h$ = 0.3) leads to the formation of spilling wave breaking over the horizontal step front and triggers, thereby pulsating wave impacts.

The horizontal distribution of vertical wave impacts (Figure 8b) shows that the maximum vertical wave impact for the small steps is located slightly below the *SWL*. The maximum vertical impact has an amplitude of about 50% of the maximum horizontal impact ($P_{max,vertical}$ = 0.5 $P_{max,horizontal}$). Note that, in the context of the onshore-orientated wave propagation and wave breaking process, this finding is reasonable for design and constructional purposes in any practical applications of stepped revetments. The vertical impact for the large steps is negligible as it represents only the hydrostatic pressure induced by the overflow of the incident wave. Additionally, data from Reference [10] are given. The relative step height of $H_{m0}/S_h$ = 7.5 is comparable to the data set provided by Reference [8] for horizontal wave impacts. The maximum presented impact is about two times larger than for the steps with a step ratio of $H_{m0}/S_h$ = 1.7. An increase in the vertical impact loads is observed for decreasing step heights and is explained analogous to the horizontal wave impact.

Figure 9 provides a non-dimensional relation between the pressure impact normalized by water density $\rho$, gravity $g$, and spectral wave height to $P_{99.6\%}/(\rho g H_{m0})$ on the abscissa and the relative position to the *SWL* ($z/H_{m0}$) on the ordinate. The probability wave impact $P_{99.6\%}$ is selected following the approach of Reference [11] to allow a comparison with the underlying data for wave impacts on vertical walls. For both examined step heights, the maximum pressure impact is close to the *SWL*. The maximum pressure decreases significantly within a range of $\pm z/H_{m0}$ around the SWL. Within a range of $\pm 2\,z/H_{m0}$, the normalized pressure impact becomes low. The $P_{99.6\%}$ wave loads measured over the stepped revetment tend to be about 50% smaller than those measured at a vertical wall. On the contrary, the impacts on the steps around the *SWL* exhibit pressures peak impacts which are comparable to those measured on vertical walls by Reference [11] (data points of single impacts not given in this figure scatter significantly along the abscissa). A dual-sided envelope curve (Equation (6)) with coefficients *a*, *b* and *c* according to Table 3 representing the best fit of data (coefficient of determination: $R^2$ = 0.62, *STD* = 0.189) is given to describe the correlation of the vertical distribution $z/H_{m0}$ of horizontal impacts $0.01 < P_{99.6\%}/(\rho g H_{m0}) \leq 3.6$.

$$\frac{P}{\rho g H_{m0}} = min\left\{ tan\left[\frac{z/H_{m0} + a}{b}\right] / c, 3.6 \right\} \tag{6}$$

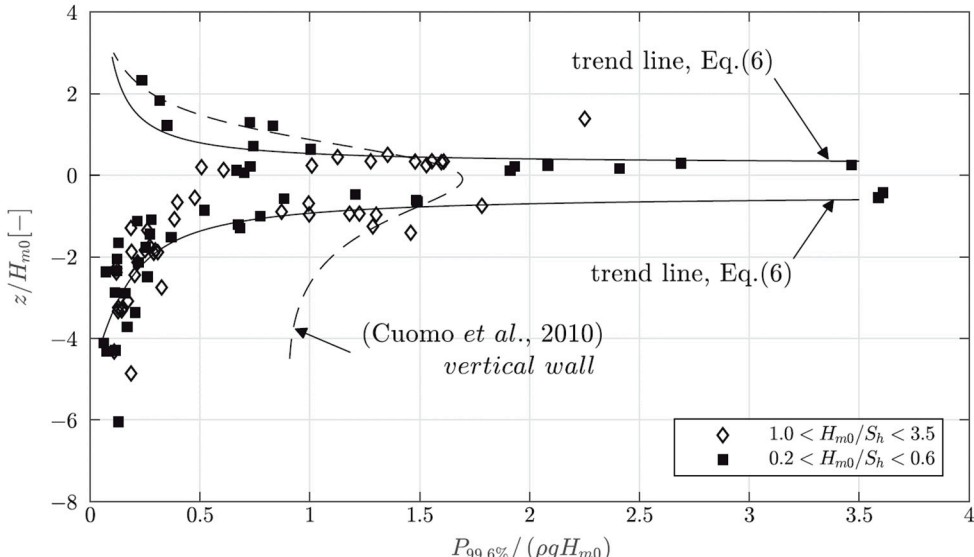

**Figure 9.** Normalized pressure impact relative to the *SWL* ($z = 0$) for a 1:2 inclined stepped revetment.

**Table 3.** Coefficients *a, b,* and *c* for Equation (6).

| z (SWL at z = 0) | a | b | c |
|---|---|---|---|
| $z \geq 0$ | −1171.64 | 745.72 | −2831.66 |
| $z < 0$ | 4.97 | −2.87 | −6.15 |

Normalized horizontal pressure impacts with varying probability of exceedance and corresponding relative water depth $z/H_{m0}$ for revetments with large ($0.2 < H_{m0}/S_h < 0.6$, Figure 10a–d) and small ($1.0 < H_{m0}/S_h < 3.5$, Figure 10e–h) steps are shown. Each row represents data with a certain probability of exceedance, as described in Section 4.1. For comparison, the reference line for the horizontal impact forces [11] and the trend line according to Equation (6) are given. The pressure distribution predicted for $P_{99.6\%}$ values by Equation (6) also shows a good agreement for $P_{max}$ values, if the step height is larger than the wave height (a). $P_{max}$ values for revetments with step heights smaller than the wave height (e) show higher impacts, which scatter in a range of $\pm 2\ z/H_{m0}$. The higher and more variable distributed impacts for small steps compared to large steps can again be ascribed to the influence of higher aeration [8,9]. With increasing probability of exceedance ((c,d) for large steps and (g–h) for small steps) the peaks around the *SWL* become less prominent, although the peaks are still visible. This trend is reasonable as the statistical distribution of the individual wave heights in a wave spectrum follows a Rayleigh distribution and the mean wave height of all waves, exceeding a certain threshold, decreases significantly with increasing probability of exceedance. Additionally, the shape of the front of a breaking wave governs the subsequent wave impact as the wave's asymmetry has a secondary importance on the resulting impact. The overall distribution over the water depth becomes more homogeneous with the decreasing probability of exceedance for reasons discussed in Section 4.2.

It has to be clarified that Equation (6) is based on experimental data by means of mathematical fitting routines. Each data-point represents the average of the four maximum impact values recorded by all implemented pressure sensors during a test. Hence, these four impacts do not necessarily correspond to or stem from the same wave or wave train. Furthermore, Equation (6) is based on a fitting through all data points. Individual under-estimations larger than the given mean standard deviation (up to factor 3) are present for single measurements. Similar observations are stated by Reference [11]. Despite these constrains, the approach is applicable to reliably estimate the location, magnitude and trend of impact pressures on stepped revetments. Further, the formula enables an estimation of the wave-induced local impacts and loads on a stepped revetment, and thus provides

the first robust estimate in the practical design and dimensioning of adequate foundations and anchoring systems.

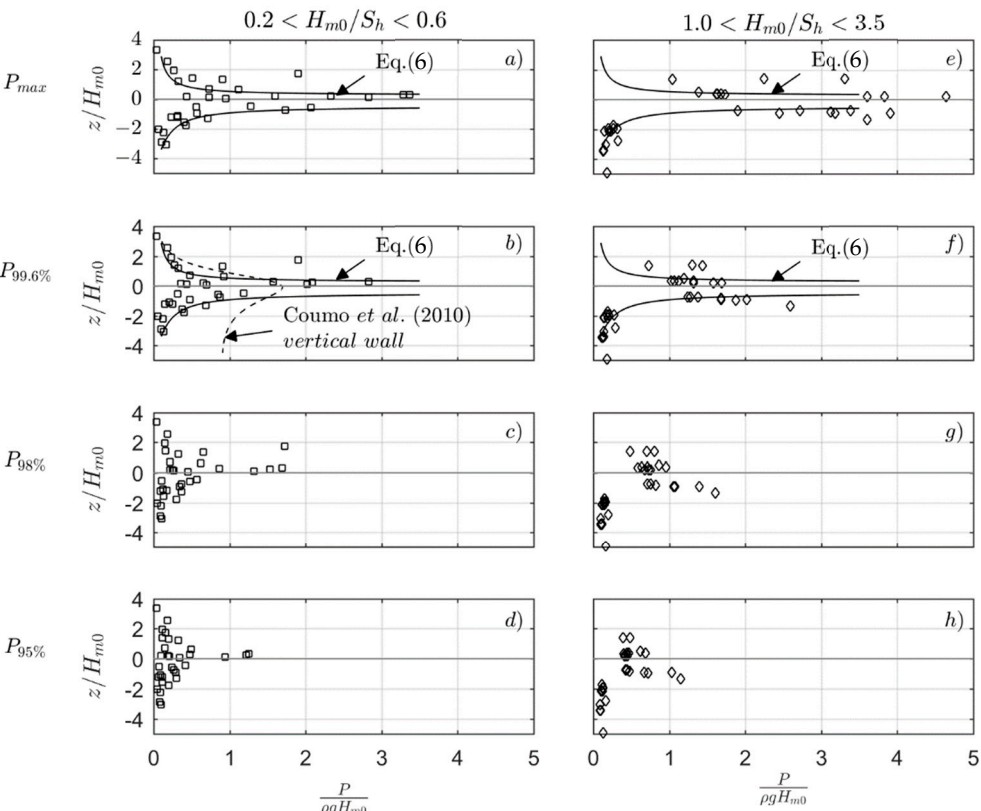

**Figure 10.** Normalized horizontal pressure impacts with varying probability of exceedance and corresponding relative water depth $z/H_{m0}$ for large ($0.2 < H_{m0}/S_h < 0.6$) and small ($1.0 < H_{m0}/S_h < 3.5$) stepped revetments including theoretical laws for stepped revetments according to Equation (6) and for vertical walls according to Reference [11].

## 5. Laboratory and Scale Effects

The tests are conducted in a 2D wave flume. The natural multi-directionality of prototype wave conditions is simplified. This affects the wave breaking and accordingly the measured impact forces. Waves are generated in intermediate water depth over a horizontal flume bottom. Hence, waves transform non-linearly during propagation onto and in interaction with the structure. Furthermore, reflected waves from the structure are re-reflected at the piston wave board as experiments were conducted without active reflection compensation. The influence of both boundary conditions on the calculated wave parameters ($H_{m0}$, $T_p$), minimized by an analysis of the incident waves very close to the structure.

Assuming gravity as dominant physical force, the model tests are Froude scaled. Therefore, inertia forces are balanced to prototype conditions whereas other physical forces as viscosity, elasticity, surface tension are incorrectly reproduced [17]. As the latter forces have an influence on the process of aeration after wave breaking and during wave run-up flow on the structure, the aeration in the Froude scaled model is reproduced incorrectly. In accordance with Reference [18], the phenomenon and process of aeration in hydraulic models, e.g., in hydraulic jumps is significant lower in smaller scales and cannot be achieved under Froude similitude. This outcome is somehow evident in any hydraulic modeling attempts and holds true for stepped revetments likewise. Hence, the presented impact pressures (in small scaled experiments) are probably over predicted as already confirmed for vertical walls by Reference [19]. As of this, an attempt of upscaling the measured impact pressures is

not taken in this paper as a detailed comparison with data from large scale model tests on stepped revetments still have to be done in due time. Accordingly, the presented data enable a discussion of the physical phenomena and triggered processes as well as relative magnitudes of impact pressures on stepped revetments and differences in trends for varying step ratios without providing exact, i.e., real-world estimates for structural design or dimensioning.

## 6. Conclusions

Wave-induced impacts on stepped revetments have been investigated by means of physical model tests conducted in a wave flume. The tests focused on the wave impact on stepped revetments with relative step heights in a range of $0.3 < H_{m0}/S_h < 3.5$.

It was found that the probability distributions of wave impact pressures on stepped revetments follow a log-normal distribution, which confirm initial findings by Reference [12,13,15]. The maximum peak amplitude in a test series decreases with increasing probability of exceedance. The decay is more intensive for step heights larger than the wave height due to significant wave transformation processes over the dominant horizontal step fronts. Large steps ($H_{m0} < S_h$) show similar load cases compared to vertical walls. For small steps ($H_{m0} > S_h$) the impact peak is also clearly visible but the subsequent quasi-static peak $P_q$ is not as prominent as for large steps or vertical walls. Therefore the recommendation of Reference [7] to calculate the forces on stepped structures with the same method as for vertical walls, which only holds true for large steps.

It was found that at small steps the highly aerated flow that emerges after the initial wave impact generally leads to smaller quasi-static peaks. The initial violent impact dissipates more energy than a pulsating load induced by wave run-up. As a result, it can be deduced that the overall energy dissipation at small step heights is larger. Impacts measured at stepped revetments below the *SWL* show relative rising times of $t_r/T_{m-10} > 0.2$. No impacting load case was detected at stepped revetments with large or small step height below the *SWL*. Real impacts are buffered by a water layer protecting the steps from violent impacts. Impacting conditions occur only very close to the *SWL*. Compared to impacts on vertical walls, the impact rising times are in the same range. The minima for stepped revetments is $t_d = t_r$. As the importance of aeration in the run-up to the wave impact is identified as a future study, it should focus on this effect in analogy to Reference [15].

The spatial distribution of the impact loads generally decrease with increasing distance to the *SWL*. The smaller the step height compared to the wave height, the higher the horizontal loads become. The reduced impact for increasing step height can be ascribed to the delayed run-down of the previous wave impact. The maximum vertical wave impact for the small steps ($H_{m0} > S_h$) is located slightly below the *SWL* with an amplitude of about 50% of the maximum horizontal impact. The vertical impact for the large steps ($H_{m0} < S_h$) is negligible as it represents only the hydrostatic pressure induced by the overflow of the incident wave. An increase in the vertical impact loads is seen for decreasing step heights analogous to the horizontal wave impact. The maximum impact decreases significantly within a range of $\pm z/H_{m0}$, mainly in the range of $\pm 2 z/H_{m0}$. The highest ($P_{99.6\%}$) wave loads measured over the stepped revetment tend to be about 50% smaller than those measured at a vertical wall. On the contrary, the impacts on the steps around the *SWL* showed impacts comparable to those measured on a vertical wall. Higher and more variable distributed impacts for small steps ($H_{m0} > S_h$) compared to large steps can be explained by the influence of higher aeration. With increasing probability of exceedance, the peaks around the *SWL* are less significant.

The analysis of the data including insights from the literature [17–19] showed the significant influence of scale effects on the results, especially in terms of the aeration. The future work will address these topics in order to enable the upscaling of the presented data. Furthermore, additional slopes have to be analyzed to determine this influence on the magnitudes of impact pressures in context of the step ratios and fully align the research in the existing findings for vertical walls and plain slopes.

**Author Contributions:** N.B.K. and To.S. conceived of the presented idea. N.B.K. developed the theory and performed the study. Ta.S. and To.S. verified the analytical methods. To.S. encouraged N.B.K. to investigate a

comparison with wave impacts on vertical walls and supervised the findings of this work. All authors discussed the results and contributed to the final manuscript.

**Funding:** The presented findings were developed within the framework of the research project 'waveSTEPS – Wave run-up and overtopping at stepped revetments' (03KIS118) funded by the Federal Ministry of Education and Research (BMBF) through the German Coastal Engineering Research Council (KFKI).

**Conflicts of Interest:** The authors declare no conflict of interest.

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
