# Peer review of "Wave Impact Pressures on Stepped Revetments"

_jmse, doi:10.3390/jmse6040156_

Round 1

Reviewer 1 Report

In general, the work presents a limited number of references and the literature review could be more updated. The structure of the manuscript could be improved by reorganizing the Introduction, including a reference to the structure of the manuscript, and transferring the section 1.1 to a new section 2, with a complete literature review.

·         Line 27/28: What is the reference stating that stepped revetments have become more attractive?

·         Line 36: As section 1.2 does not exist, there is no need to have a section 1.1! Section 1.1 should be named as a new section 2, with a deeper and more recent literature review. Introduction section may be complemented with information about the manuscript structure and explicit description of the research goals.

·         Line 87: A width of 2.2 meters is referred but Figure 2a presents 0.7 meters. Please, make it clear.

·         Line 91: The length of the flume is not shown in Figure 1. The equipment refereed in lines 92 to 96 is not presented in Figure 1. A new figure may be added to the manuscript to represent the complete longitudinal section of the flume, helping on understanding all the relative positions of the equipment in the laboratory.

·         Line 115: Has hs in the text the same meaning of d, in figure 1? Please, clarify this.

·         Line 137: Results section is in fact Results and Discussion and should be renamed in accordance.

·         Lines 140-142: The recent literature here referred is not described in the literature review, at the beginning (lines 37 to 84).

·         Line 149: Hm0 of 0.08 m is not completely in accordance with Table 1 (0.084 or 0.089). The same precision should be considered for the values in the Table and in the text. The same for the Iribarren number (3.0, or 2.8 and 2.9), on line 150. This is valid for all the document.

·         Line 152: Which is the correct maximum pressure? Is it 1.25 kPa or 1.26 kPa, as presented in Figure 3a?

·         Line 157: The quality of the figures should be improved, avoiding the over position of the numbers, at the vertical axis.

·         Line 159: Is it P1 or P2?... line 151 refers P2!

·         Lines 197 to 200: The values do not completely match with the ones presented in Table 1.

·         Line 206: The value presented at Figure 5g (1.61) should be 1.68 kPa, to be in accordance with Figure 3b.

·         Line 208: The different figures along Figure 5 are not identified in the figure.

·         Line 260: The reference of Kortenhaus et al. (1999), presented in Figure 6, is not listed in the References.

·         Equation 1: where is tt, should be tr!

·         Line 292: If references presented in Figure 7 correspond to the ones listed in the References list, “et al” is missing, both for Heimbaught (1988) and Melby (2009).

·         Line 294: Values of 0.08 and 3.0 in the figure caption should be corrected in accordance with Table 1.

·         Line 305: Equation number presents an error! The end of the line is “t” instead of “the”!

·         Line 310: Equation number presents an error!

·         Lines 315 to 328: the text should be presented before Figure 9.

·         Lines 319 and 321: Equation number presents an error!

·         Line 341: the word “only” is repeated. Delete the second “only”.

·         Line 349: Replace “rage” by “range”.

References number 5 and 8 should be more complete.

Author Response

The authors like to thank the reviewer to pay attention to the paper and its content. The given comments and suggestions were each for itself helpful and reasonable. We appreciate the reviewer’s efforts by giving answers to all questions and include all comments to the final paper. Comments to each individual question are given after each item in this document and are marked in red. The revisions have been included to the paper using the track changes function, which should simplify the identification of changes to the three reviewers.

Detailed comments are given in the attached file.

Reviewer 2 Report

The present manuscript describes the pressure field induced by spectral waves over stepped revetments. Laboratory experiments have been carried out by testing different step geometries, wave characteristics and water depths. Results on measured horizontal and vertical pressure distributions, impact duration and rising time, and comparisons with the vertical wall case are shown.

The manuscript is well written and organized, with good comparisons with theory and existing data.

Some minor errors should be amended to grant publication.

Specific aspects

The use of stepped revetments is strongly connected with the hydrodynamics deriving from the wave impact, especially for what concerns engineering perspectives. Hence, references to the typical reflection problems (e.g., which may occur in harbors, hence destabilizing moored vessels) or erosion issues (scour under the toe of the structure, this often leading to structure damage or failure) should be included (e.g., see Yin et al., 2017; Postacchini et al., 2016; Sumer et al., 2005).

At the end of the introduction (L84), please include the description of the paper sections.

In the presentation of the results, a better description of Führböter [11]’s findings need to be included, to properly understand how good the comparison is with what illustrated in the manuscript (L176-177).

In the description of Fig.6, some considerations on the goodness of fit or the adaptation of the best-fit lines to the data should be added. Further, the caption should include the description of all lines reported in Fig.6.

Similarly, the caption of Fig.7 should also include the theoretical laws reported in the figure for comparison purposes.

Specific points

·        L48: what do the authors mean with “this” (this study or spilling breakers)?

·        L49-52: about the described jet dynamics, a suitable reference should be included.

·        L115-116: “intermediate water depths” seems not to be formally correct, please rephrase.

·        L133: the authors should try to include the number of waves tested during their experiments.

·        L134: why “certain”? it should be 99.6%.

·        L149-150: Iribarren number and wave height for tests 103 and 209 do not correspond to those presented in table 1.

·        L155-156: the sentence “Slight deviations …” is not clear and should be reworded.

·        L197: what do the authors mean with “0.03 in”?

·        L246 and L251: please remove the unnecessary brackets.

·        L274-277: sentences are here quite awkward, please reword.

·        L288: do the authors mean 7.5 instead of 11?

·        L305 to L321: please amend all reference-source errors.

·        L338: it should be “clearly visible”.

References

Yin, Z., Jin, L., Liang, B., & Wang, Y. (2017). Numerical Investigation of Wave Reflection from a Stepped Breakwater. Journal of Coastal Research, 33(6), 1467-1473.

Postacchini, M., Russo, A., Carniel, S., & Brocchini, M. (2016). Assessing the hydro-morphodynamic response of a beach protected by detached, impermeable, submerged breakwaters: a numerical approach. Journal of Coastal Research, 32(3), 590-602.

Sumer, B. M., Fredsøe, J., Lamberti, A., Zanuttigh, B., Dixen, M., Gislason, K., & Di Penta, A. F. (2005). Local scour at roundhead and along the trunk of low crested structures. Coastal Engineering, 52(10-11), 995-1025.

Author Response

The authors like to thank the reviewer to pay attention to the paper and its content. The given comments and suggestions were each for itself helpful and reasonable. We appreciate the reviewer’s efforts by giving answers to all questions and include all comments to the final paper. Comments to each individual question are given after each item in this document and are marked in red. The revisions have been included to the paper using the track changes function, which should simplify the identification of changes to the three reviewers.

Detailed answers are provided in the attached file.

Reviewer 3 Report

Comments attached.

Author Response

The authors like to thank the reviewer to pay intense attention to the paper and its content. The given comments and suggestions were each for itself helpful and reasonable. We appreciate the reviewer’s efforts by giving answers to all questions and include all comments to the final paper. Comments to each individual question are given after each item in this document and are marked in red. The revisions have been included to the paper using the track changes function, which should simplify the identification of changes to the three reviewers.

Detailed answers are given in the attached file.
